# Improving the Autogenous Self-Sealing of Mortar: Influence of Curing Condition

**DOI:** 10.3390/ma14082068

**Published:** 2021-04-20

**Authors:** Lianwang Yuan, Min Li, Yongbo Huang, Zonghui Zhou, Congqi Luan, Zunchao Ren, Yongyi Liu, Tongtong Zhou, Xin Cheng, Jinbang Wang

**Affiliations:** 1Shandong Provincial Key Laboratory of Preparation and Measurement of Building Material, University of Jinan, Jinan 250022, China; lwyuan0910@126.com (L.Y.); mse_huangyb@ujn.edu.cn (Y.H.); luancq0822@163.com (C.L.); ren295274426@163.com (Z.R.); lyy965335812@163.com (Y.L.); mse_chengx@ujn.edu.cn (X.C.); 2School of Materials Science and Engineering, University of Jinan, Jinan 250022, China; 3Shandong Sanjian Group Co., Ltd., Jinan 250100, China; 15562486005@163.com; 4Shandong Hi-Speed Road & Bridge Maintenance, Ltd., Jinan 250100, China; zhou_tongtong@foxmail.com

**Keywords:** autogenous self-sealing, curing condition, relative humidity, state of water, continuous hydration, carbonation

## Abstract

With the construction of projects under severe environments, new and higher requirements are put forward for the properties of concrete, especially the autogenous self-sealing property, which is greatly affected by the curing environment and the state of the water. Herein, six types of curing conditions, including in air with a relative humidity of 30%, 60%, and 95%; flowing water; wet-dry cycles; and static water, are designed to investigate the autogenous self-sealing of mortar under different curing conditions. The results showed that the self-sealing ratios are higher than 60% and the cracks are closed for the mortar undergoing the wet-dry cycles and the static water. However, the self-sealing ratios of mortar are lower than 10% and the cracks are almost unchanged when the mortar is cured in the air with a relative humidity (RH) of 30% and 60%. The static liquid water is more conducive to the continued hydration of cement and the formation of CaCO_3_ than the flowing water. The research provides guidance for the design of concrete and the improvement of autogenous self-sealing when the concrete serves in different environments.

## 1. Introduction

With the construction of some major projects related to national economy and important projects under severe conditions, new and higher requirements are constantly put forward for the properties of cement-based materials, such as high autogenous self-healing performance when micro-cracks tend to form under severe conditions. Therefore, how to improve the self-sealing property of cement-based materials has been attracting more and more attention.

As is known, the autonomous self-healing of concrete could be achieved by adding foreign aid materials. Roig-Flores et al. [1] found that the self-healing ability of concrete was enhanced with the addition of crystallization. Feiteira et al. [2] used the encapsulation of repair agents to achieve effective healing of cracks. Furthermore, bacteria-based microcapsules [3,4] and super absorbent polymers [5] were also used to promote the autonomous self-healing of concrete. It is concluded that the self-healing of cement-based materials is affected by foreign substance. However, the addition of foreign aid materials brings the extra cost of concrete and the easy failure of these materials in harsh environments restricts their promotion and application. Therefore, the improvement of autogenous self-healing is an important research direction for cement-based materials.

Autogenous self-sealing is an inherent characteristic of cement-based materials, which is regarded as a kind of self-healing performance and also an important performance of the environmental adaptability. Under suitable conditions, the micro-cracks of cement-based materials can be sealed autogenously by the continued hydration of unhydrated cement particles and the carbonation of hydration products [6].

Some interesting and meaningful studies have been carried out to promote the autogenous self-sealing properties of concrete. For example, coarse cement particles were added into concrete to enhance the continued hydration [7]. Yuan et al. [8] found that cement particles with the size of 30–60 μm contribute to the micro-cracks self-sealing of concrete at the hydration age of 28 days. Qiu and Huang et al. [9,10] demonstrated that self-healing ability was improved for concrete containing supplementary cementitious materials.

The major mechanisms of autogenous self-sealing are the continued hydration of unhydrated cement particles and the carbonation of hydration products. Therefore, autogenous self-sealing is greatly affected by the curing conditions [11], especially the state of water. For the curing condition of liquid water, the unhydrated cement particles are easy to hydrate and the CaCO_3_ crystals are easy to grow. Additionally, water could also promote self-healing for the concrete with crystalline admixtures [1,12]. Qian et al. [13] found that concrete containing bacteria exhibited a better self-healing behavior when it is submerged in water. Therefore, water is dispensable for the self-healing of cement-based materials.

The service environments of cement-based materials are not only limited to liquid water: most of them serve in air with different relative humidity (RH). Cement-based materials in plateaus and deserts work in dry air with an RH of 20~40%, such as Lhasa. The RH of Beijing and Xining is around 60%, where a mild air is provided for cement-based materials. Humid air with an RH of 75~85% is provided for cement-based materials, such as in Guangzhou and Shanghai, and the RH can reach up to 98% in the eve of a rainy day. Additionally, some cement-based materials undergo flowing water and the dry-wet cycles. The rainfall precipitation provides the flowing water condition for cement-based materials, while the cement-based materials at coastlines and splash zones work in a wet-dry cycles condition because of the tidal action. Meanwhile, many cement-based materials serve in a static water condition, such as dams and underground pipe galleries.

The purpose of this study is to investigate the effect of curing conditions, especially the state of water, on the autogenous self-sealing of cement-based materials. Therefore, curing conditions, including air with an RH of 30%, 60%, and 95%; flowing water; wet-dry cycles; and static water, were designed to study the self-sealing of mortar. The self-sealing of the surface cracks was evaluated by the changes of the crack width and the water permeability. The cracks inside the mortar were assessed by ultrasonic tests with the parameters of ultrasonic pulse velocity (UPV), head wave, amplitude, and frequency. This research will provide guidance for the design of concrete and the improvement of autogenous self-sealing when the cement-based materials serve in different environments.

## 2. Materials and Methods

### 2.1. Materials and Samples Preparation

The 42.5 ordinary Portland cement (OPC) [14,15] was used, with the chemical composition and physical properties shown in Table 1 and Table 2, respectively. River sand with a maximum particle size of 4.75 mm was used as the fine aggregate. The mass percentages of sand particles in the range of 0.08~0.15 mm, 0.15~0.3 mm, 0.3~0.6 mm, 0.6~1.18 mm, 1.18~2.36 mm, and 2.36~4.75 mm were about 11% and 21%, 27%, 24%, 12%, and 5%, respectively. The fineness modulus of the river sand was 2.8. The water to cement ratio was 0.4 and the cement to sand ratio was 1:3.

The mortar was prepared by stirring the mixture at a speed of 62 R/min for 120 s and 125 R/min for 60 s. Then, the mortars were cast in the mold with the dimension of 40 mm × 40 mm × 160 mm, 50 mm × 50 mm × 50 mm, and φ100 × 50 mm, respectively. Prisms were used for the observation of crack width and the UPV test, the cubes were used for the ultrasonic test, and the cylinders were used for the water permeability test. After curing in a standard chamber with RH of 95% and temperature of 25 ± 2 °C for 24 h, the mortars were demolded and cured in the standard chamber (25 ± 2 °C, 95% RH) for 28 days.

### 2.2. Self-Sealing Conditions Design

According to the different service environments for the mortar, six self-sealing conditions were designed, including dry air with an RH of 30% (DC), mild air with an RH of 60% (MC), humid air with an RH of 95% (HC), flowing water (FW), wet-dry cycles (WD), and static water (SW). Accordingly, such conditions were simulated through a constant temperature and humidity curing container (RH of 30%, 25 °C), an outdoor environment (RH of 60%, 25 °C), a standard curing room (RH of 95%, 25 °C), a simulated rainfall device (water flowing rate of 200 mL/min, 25 °C), a wet-dry cycle curing tank (a cycle of drying for 12 h and wetting for 12 h, 25 °C), and a water tank (immersion depth of 50 mm, 25 °C), respectively. The self-sealing behaviors of mortars in these conditions were studied. It should be noted that the actual RH of outdoor environment was around 55%~65% and the temperature was 25 ± 4 °C during the test.

### 2.3. Experimental Procedures

#### 2.3.1. Preparation of a Single Crack

A single crack with the width about 180~210 μm for hardened mortar was prepared. A prepared gap served as the start of the crack. The prisms were cracked under a three-point bending procedure with a loading rate of 100 N/s [16]. The cracked prisms were fixed with epoxy resin and acrylic board to ensure that the cracks did not change due to external forces, as shown in Figure 1a. Note that the epoxy resin must not flow into the cracks. The prisms were used for the crack observation and UPV test. Other single cracks in cylinders and tubes were pre-cracked under the split test with a loading rate of 100 N/s. The slow split loading cannot break the cylinders and tubes into two parts. Moreover, to prevent external forces from changing the crack width during the continued curing and testing process, the epoxy resin was required to be fixed on the side of the cylinders and the top and bottom of the cubes. The cylinders were used for the water permeability test and the cubes were used for the ultrasonic waveform test. The preparation methods of the cracks are shown in Figure 1b,c.

#### 2.3.2. Evaluation for the Closure of Surface Cracks

The surface crack was observed by the crack width gauge (PTS-C10, Wuhan Botest Instrument co., LTD, Wuhan, China) and the width was recorded to evaluate the closure of the surface cracks. A position with the crack width around 180~210 μm was found and labeled in each crack. When the samples were cured in different conditions for 60 days, the crack width of the labeled position was tested again.

#### 2.3.3. Water Permeability

The water permeability test gives accurate measurements about the regaining of liquid tightness over time. The schematic diagram of the water permeability test is shown in Figure 2. At the self-sealing age of 7, 14, 28, 60, 90, and 120 days, the weight of permeated tap water was recorded and the water permeability (Q) could be calculated by Equation (1):*Q* = (*M_h_*/*M_c_*) × 100%(1)
where *M_c_* is the mass of passed water per minute of the sample after making the crack, and *M_h_* is the mass of passed water per minute of the same sample after the curing period for the self-sealing process.

#### 2.3.4. Ultrasonic Tests

An ultrasonic non-destructive test is frequently used for assessing the cracks of concrete [17]. UPV is considered to be a reliable detection parameter of ultrasonic tests for evaluating the damage and self-healing of concrete [18,19,20]. The TICO ultrasonic instrument (Shanghai Lrel Instrument co., LTD, Shanghai, China) and mechanism for the UPV test are shown in Figure 3. The samples were dried at 50 °C for 24 h before the UPV test to eliminate the influence of free water on the ultrasonic transmission. Additionally, the energy and frequency of ultrasonic wave have also been applied to evaluate the damage and repair of concrete [8,21,22]. The principle of the transmitted ultrasonic wave and the equipment are shown in Figure 4. The ultrasonic waveform was also converted from the time domain to the frequency domain by the fast Fourier transform (FFT).

#### 2.3.5. Characterization of Self-Sealing Products

The self-sealing products were scraped from the cracks when the samples cured in different conditions for 60 days. The mineral compositions of the self-sealing products were characterized by X-ray diffraction (XRD, Bruker D8 Advance, Karlsruhe, Germany) analysis within the 2θ scanning range of 5–80° with a step size of 0.02°.

Thermogravimetric analysis (TGA, Mettler Toledo, Shanghai, China) was used to determine the mineral content of the self-sealing products. Approximately 30 mg of self-sealing products were placed in an alumina crucible and heated from 25 °C to 1000 °C at 10 °C/min. An argon atmosphere was used for protection.

Scanning electron microscope (SEM, Zeiss, Jena, Germany) was used to observe the microscopic morphology of self-sealing products produced in different surroundings. After being dried at 50 °C for 24 h, the sample surfaces with self-sealing products were covered with gold to satisfy the requirement of conductivity.

## 3. Results and Discussion

### 3.1. Closure of Surface Crack

The measurement of surface crack width is one of the most intuitive methods for evaluating the self-sealing of concrete. Figure 5 shows the closure of surface cracks for the mortars cured in different conditions over 60 days. It can be seen that the excessive self-sealing products fill in the cracks for the samples experiencing the wet-dry cycles and the static water. Thereby, the cracks are completely closed. The static liquid water is beneficial to the formation of the self-sealing products. However, the flowing water takes away the Ca^2+^ and weakens the precipitation of CaCO_3_, which presents a crack width of 93 μm reduction value for the samples cured in the flowing water. Compared with the static water, the flowing water is not conducive to the self-sealing of the mortar. The values of crack width decrease for the samples cured in the air with the RH of 30%, 60%, and 95% are 2 μm, 21 μm, and 165 μm, respectively. For the mortar self-sealed in air, the good self-sealing behavior happens easily in the air with an RH of 95% and above. This is because the air with high RH is more likely to condense into liquid water.

### 3.2. Water Permeability

The self-sealing of surface cracks can also be evaluated by the water permeability. Figure 6 shows the water permeability of the samples cured in different conditions. As the cracks closed, the water permeability decreased. For the samples cured in the dry air with 30% RH and the mild air with 60% RH, there was merely a little change in the water permeability until the self-sealing age of 120 days, which was still higher than 95%. Compared with the former, the water permeability dropped to 53% for the samples cured in the humid air at the self-sealing age of 60 days. When the samples continued to self-seal in this condition, the water permeability was almost unchanged. A similar change of the water permeability for samples cured in flowing water was obtained. The water permeability of samples cured in flowing water is slightly higher than that in the humid air. For the samples cured in the wet-dry cycles and the static water, the water permeability decreased rapidly at the early age of self-sealing and dropped to 0 at the self-sealing age of 60 days. Therefore, the continuous curing in static water is beneficial to the self-sealing of the mortar because of the leaching of Ca^2+^ and the precipitation of CaCO_3_ [18]. With the prolonging of the self-sealing age, the crack surfaces are covered by the self-sealing products and the transmission of ions (Ca^2+^, CO^3−^, et.) is blocked [23]. Therefore, the continued hydration and carbonation are inhibited, which is the reason for the little change of water permeability in the later age of self-sealing. Moreover, at the same curing age, the water permeability decreased more sharply in static water than in wet-dry cycles, which indicated that the rate of self-healing in static water is faster than that in wet-dry cycles. The samples immersed in water were in contact with the water for a longer time than those cured in wet-dry cycles. With the extension of immersion time in water, the continued hydration degree of cement particles increased, and furthermore the carbonation degree of hydration products and the production of CaCO_3_ increased [6,11].

### 3.3. Ultrasonic Tests

#### 3.3.1. UPV

When the transmission path of an ultrasonic wave is changed because of the different crack depth and width, the UPV values are also different. The results of the UPV test are shown in Figure 7.

The UPVs of the uncracked samples cured for 28 days in the standard curing condition and the cracked samples were about 4000 m/s and 1000 m/s, respectively. After curing over 60 days for the self-sealed samples, the UPVs fell within the range of 1000 m/s–4000 m/s and increased the improvement of self-sealing ability. For the samples self-sealed in the dry air, mild air, humid air, flowing water, wet-dry cycles, and static water, the UPVs were about 1280 m/s, 1390 m/s, 2150 m/s, 2010 m/s, 2870 m/s, and 3080 m/s, respectively. Based on the principle of shortest transmission path, the higher value of UPV shows a better self-sealing ability of mortar. Therefore, the water promotes the self-sealing of the mortar. Additionally, the UPV of the self-sealed samples with the complete closure of cracks was still lower than that of the uncracked samples. On the one hand, the self-sealing products are looser than the matrix of the mortar. On the other hand, it is also considered that even if the crack is filled at the surface the interior of the crack maybe not completely filled with sealing products [17].

#### 3.3.2. Head Wave of Ultrasound

When the ultrasonic wave transmits though the mortar, the inside cracks weaken and hinder the transmission, which can be measured by the arrival time and amplitude of the head wave [24]. Figure 8 and Figure 9 show the self-sealing results based on the head wave of ultrasound for the samples cured in different conditions.

The arrival times of the head wave for the uncracked samples were about 14 μs. At the same time, their amplitudes are between 3~5 mV. For the cracked samples, the arrival times of the head wave extended to 34 μs and the amplitudes dropped to ~1 mV. With the self-sealing of samples, the arrival time of the head wave was shortened and the amplitude increased compared with the cracked samples. In terms of the arrival time of the head wave, the following sequence was observed: static water (13 μs) < wet-dry cycles (15 μs) < humid air (22 μs) < flowing water (26 μs) < mild air (31 μs) < dry air (34 μs). At the same time, in terms of the amplitude of the head wave, the following sequence was also observed: static water (~4.7 mV) > wet-dry cycles (~3.4 mV) > humid air (~2.6 mV) > flowing water (~2.5 mV) > mild air (~1.9 mV) > dry air (~0.9 mV). The shorter arrival time and the greater amplitude demonstrate a better self-sealing ability of mortars. Therefore, the static liquid water is also conducive to the self-sealing of cracks inside the mortar.

To study the self-sealing of mortar, the self-sealing ratios are calculated by two methods of the arrival time (*E_t_*) and the amplitude (*E_a_*) of head wave though Equations (2) and (3), respectively:*E_t_* = (*T_h_ − T_c_*)/(*T_u_ − T_c_*) × 100%(2)
*E_a_* = (*A_h_ − A_c_*)/(*A_u_ − A_c_*) × 100%(3)
where *T_u_*, *T_c_*, and *T_h_* are the arrival times of head waves for the uncracked, cracked, and sealed samples, respectively. *A_u_*, *A_c_*, and *A_h_* are the amplitudes of head waves for the uncracked, cracked, and sealed samples, respectively.

The results of the self-sealing ratios are shown in Figure 9. The *E_t_* and *E_a_* of the samples experiencing the static water were close to 100%. The best self-sealing of mortar depended on the continuous hydration of cement particles and the carbonization of hydration products, which benefit from the static liquid water. For the samples cured in the air with 30% RH, the *E_t_* and *E_a_* were below 10%. Therefore, the self-sealing of mortar is difficult to improve in dry air. Although the liquid water is indispensable for the good self-sealing of mortar, the *E_t_* and *E_a_* of the samples cured in the flowing water are lower than that of the samples cured in the air with 95% RH. This is because the Ca^2+^ is taken away by the flowing water, which thereby adversely affects the formation of CaCO_3_.

#### 3.3.3. Ultrasonic Waveform

Ultrasonic energy will be weakened through the absorption and scattering of the aggregates, pores, and cracks when the ultrasonic waves transmit in concrete [25]. The energy loss is characterized by the maximum amplitude reduction (*ΔV*) of ultrasonic waveform [22]. Figure 10 and Figure 11 show the ultrasonic waveform of the samples and the changes of the maximum amplitude.

The maximum amplitudes of ultrasonic waveform are close to ~30 mV for the uncracked samples. When the samples cracked, the maximum amplitudes decreased to ~5 mV. With the self-sealing of the cracks, the maximum amplitude increased and the energy loss of ultrasound decreased. In terms of the maximum amplitude of the sealed samples cured in different conditions, the following sequence was observed: static water (25.9 mV) > wet-dry cycles (23.4 mV) > humid air (17.6 mV) > flowing water (11.5 mV) > mild air (7.6 mV) > dry air (5.3 mV). For the samples cured in liquid water, a good self-sealing behavior was obtained due to the continuous hydration and the formation of CaCO_3_. The self-sealing of mortar cured in the flowing water was not as good as that cured in the static water, which is due to the loss of Ca^2+^. The self-sealing of mortar is difficult to achieve in the samples cured in air with 30% RH and 60% RH, but the samples cured in the air with 95% RH had a better self-sealing behavior. A possible reason is that the air with 95% RH and above can condense into liquid water on the crack surface.

To quantify the self-sealing of the mortars by ultrasound energy, the self-sealing ratios (*E_V_*) are calculated by the maximum amplitude of ultrasonic wave though Equation (4):*E_V_*= (∆*V_H_ −* ∆*V_C_*)/(∆*V_U_*− ∆*V_C_*) × 100%(4)
where Δ*V_U_,* Δ*V_C_*, and Δ*V_H_* are the maximum amplitudes of ultrasonic waves for the uncracked, cracked, and sealed samples, respectively.

It can be seen from Figure 11 that the *E_V_* values of the samples cured in the dry air with 30% RH and the mild air with 60% RH are below 10%. At the same time, the *E_V_* values of the samples cured in the static water and wet-dry cycles are above 70%. Therefore, the liquid water is more important than the air for the self-sealing of the mortar. However, the *E_V_* value of the samples cured in the humid air is higher than that in the flowing water. The same trend of self-sealing behavior as the head wave of ultrasonic wave is obtained. Therefore, it is the static water, rather than the flowing water, which plays a significant role in the self-sealing of the mortar. For the mortar serving in air, the humid air with 95% RH and above is beneficial for the self-sealing of cracks.

#### 3.3.4. Frequency Analysis of Ultrasonic Wave

The ultrasonic wave with high frequency is easy to attenuate [26] and the amplitude of dominant frequency decreases when the ultrasonic wave transmits in the cracked samples [25]. Figure 12 and Figure 13 show the amplitudes of dominant frequency for the samples with different states.

The dominant frequencies of the uncracked samples were in the range of 40~55 kHz and the amplitudes of dominant frequency were about 1.0~1.2 mV. For the cracked samples, the amplitudes of dominant frequency were below 0.4 mV. The decrease of amplitude indicates that the cracks caused a serious attenuation of ultrasonic wave. As the cracks sealed, the attenuation of ultrasonic wave was weakened, which shows the increase of the amplitude of dominant frequency. In terms of the amplitudes of dominant frequency for the samples self-sealed in different conditions, the following sequence was observed: static water (0.81 mV) > wet-dry cycles (0.75 mV) > humid air (0.58 mV) > flowing water (0.51 mV) > mild air (0.43 mV) > dry air (0.41 mV). The great amplitude of dominant frequency shows a good self-sealing behavior of the mortar cured in water. The results are similar to that of the head wave and ultrasonic waveform.

The self-sealing ratios (*E_A_*) based on the amplitude of dominant frequency are calculated by Equation (5):*E_A_*= (∆*A_H_ −* ∆*A_C_*)/(∆*A_U_ −* ∆*A_C_*) × 100%(5)
where Δ*A_U_*, Δ*A_C_*, and Δ*A_H_* are the amplitudes of dominant frequency for the uncracked, cracked, and self-sealed samples, respectively.

Figure 13 shows the results of *E_A_*. For the samples which experienced the static water and wet-dry cycles, the *E_A_* was above 60%. At the same time, the *E_A_* of the samples self-sealed in the dry air with 30% RH and the mild air with 60% RH was less than 10%. The *E_A_* of the samples cured in the humid air with 28% was higher than that of samples cured in the flowing water with 20%. The high *E_A_* shows a good self-sealing ability of the mortar. It also demonstrates that the static water promotes the self-sealing of the mortar. At the same time, the humid air is beneficial for the self-sealing of the mortar working in air.

### 3.4. Characterization of Self-Sealing Products

#### 3.4.1. XRD Analysis

XRD analysis was performed to determine the mineral composition of the self-sealing products, and the patterns are shown in Figure 14. The CaCO_3_ is detected as the dominant self-sealing product and it is mainly presented as calcite, which is consistent with the result of previous research [27]. With the increase of relative humidity, the peaks intensity of calcite increases. Moreover, the peaks intensities for samples cured in static water and wet-dry cycles were higher than other samples. This is probably attributable to the CaCO_3_ crystal produced. The production of calcite is due to carbonation caused by the calcium hydroxide dissolved in water [28].

#### 3.4.2. TG Analysis

TG analysis is used to evaluate the content of self-sealing products generated in different conditions. The main self-sealing product is CaCO_3_, which is decomposed at 630~800 °C [29]. Figure 15 shows the results of TG analysis.

Based on the weight loss, the amount of CaCO_3_ in the self-sealing products was calculated. The amount of CaCO_3_ in the self-sealing products produced in different conditions is in the following sequence: static water (21.79%) > wet-dry cycles (21.26%) > humid air (16.93%) > flowing water (16.60%) > mild air (15.54%) > dry air (15.01%). In general, the amount of CaCO_3_ produced in liquid water is higher than that produced in air. Therefore, much CaCO_3_ fills in the cracks and the self-sealing of the mortar was improved when the samples were cured in water. This is because that the Ca^2+^ in matrix is easily leached out and reacts with the HCO_3_^−^ / CO_3_^2−^ dissolved in water [23,30]. The amount of CaCO_3_ produced in the flowing water is lower than that produced in the humid air with 95% RH. This is due to Ca^2+^ being taken away by the flowing water. There are different crystal forms of CaCO_3_ in self-sealing products: amorphous CaCO_3_, aragonite, and calcite. The endothermic peak at 630–720 °C in the DTG curves represents the decomposition of aragonite [31]. At the same time, the endothermic peak at 720–800 °C is attributed to the decomposition of calcite [8,10]. Therefore, the liquid water promotes the formation of calcite, which is beneficial to the self-sealing of the mortar [28].

### 3.5. Mechanism Analysis of Autogenous Self-Sealing

SEM-EDS analysis is used to observe the microscopic morphology of the self-sealing products generated in different conditions. The results are shown in Figure 16. Moreover, Figure 17 shows the self-sealing mechanisms of the mortar cured in different conditions.

There was no typical CaCO_3_ crystal found on the crack for the samples cured in the dry air with 30% RH and the mild air with 60% RH from Figure 16 and Figure 17a. It is because that the CO_2_ is difficult to dissolve and form CO_3_^2−^ and HCO_3_^−^, which is crucial for the CaCO_3_ precipitation [32]. There are also some CaCO_3_ in the self-sealing products, known from the results of XRD and TG analysis. It is due to the carbonization of Ca(OH)_2_ on the crack surface. However, it is not enough to close the cracks, which presents a poor self-sealing ability of the mortar.

From Figure 16 and Figure 17b, the tiny crystal of CaCO_3_ and fibrous product are found on the cracks’ surface for the samples cured in the humid air with 95% RH (an actual RH of 98% was measured around the samples). Therefore, the carbonization was accelerated and the continued hydration of unhydrated cement particles occurred in the humid air [33,34]. One possible reason is that the air is condensed in the capillary pores and a layer of liquid water is formed on the crack’s surface when the RH of humid air is higher than 95%. Moreover, there is sufficient CO_2_ dissolved in the water, which is beneficial to the carbonization. Therefore, the humid air with 95% RH and above promotes the self-sealing of cracks.

The crack surfaces were covered by the dense CaCO_3_ when the cracked samples experienced the flowing water, as shown in Figure 17c. The main self-sealing product was calcite. However, the self-sealing behavior was not good. On the one hand, the continue hydration products were dissolved by flowing water. On the other hand, the loss of Ca^2+^ produced by the dissolution of Ca(OH)_2_ is not conducive to the precipitation of CaCO_3_. Furthermore, the dense calcite blocked the leaching of Ca^2+^ [35].

For the samples which experienced the wet-dry cycles and the static water, the leaching of Ca^2+^ was ongoing in static liquid water, as shown in Figure 17d. Additionally, the dissolution of CO_2_ in water was in dynamic equilibrium, especially near the liquid level. The sufficient Ca^2+^, CO_3_^2−^, and HCO_3_^−^ promoted the CaCO_3_ precipitation, which presented the increase of the amount of CaCO_3_ and the crystal size growth [36]. As a result, the static liquid water played a critical role in self-sealing of the mortar [37]. It must be emphasized that the samples are immersed in shallow water (the immersion depth of 50 mm).

## 4. Conclusions

Based on the results obtained above, the specific conclusions are presented as follows:(1)The self-sealing of concrete cracks includes the surface cracks sealing and the internal cracks sealing. The research shows that the crack width observation and the water permeability test are suitable for the evaluation of self-sealing for surface cracks and the ultrasonic tests can characterize the self-sealing of internal cracks.(2)Water promotes the self-sealing of surface cracks. The surface cracks of samples cured in static water and wet-dry cycles are closed and the water permeability of these samples is close to zero. With the increase of relative humidity, the crack width and the water permeability decrease.(3)Water contributes to the self-sealing of internal cracks. The self-sealing ratios based on the ultrasonic test for the samples cured in static water and wet-dry cycles are greater than 60%, which are higher than the samples cured in other conditions. Similarly, the self-sealing ratios also improve with the increase of relative humidity.(4)The static water promotes the self-sealing of concrete cracks, because of the acceleration of the continued hydration of unhydrated cement particles and the precipitation of CaCO_3_. The flowing water takes away the Ca^2+^ from the dissolution of the unhydrated cement particles and the hydration products, which hinders the precipitation of CaCO_3_ and thereby negatively affects the self-sealing of the cracks.

## Figures and Tables

**Figure 1 materials-14-02068-f001:**
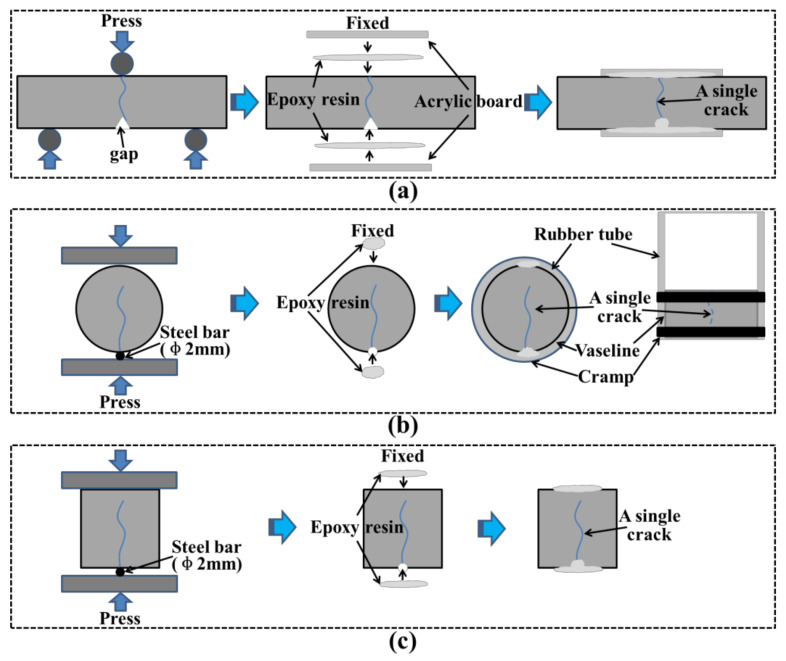
Schematic diagram of a single crack preparation: (**a**) a single crack prepared in prisms by three-point bending procedure, (**b**) a single crack prepared in cylinders by split test procedure, (**c**) a single crack prepared in cubes by split test procedure.

**Figure 2 materials-14-02068-f002:**
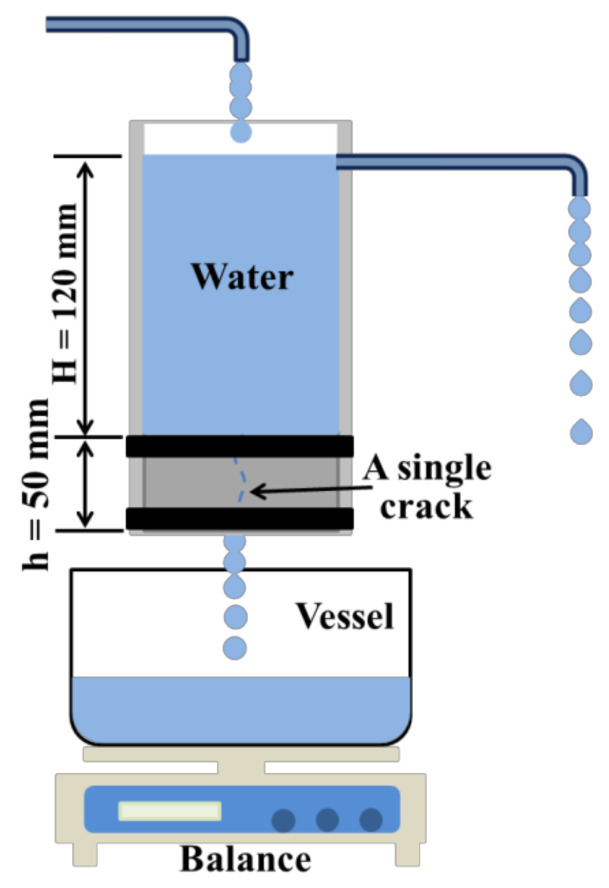
Schematic diagram of water permeability test.

**Figure 3 materials-14-02068-f003:**
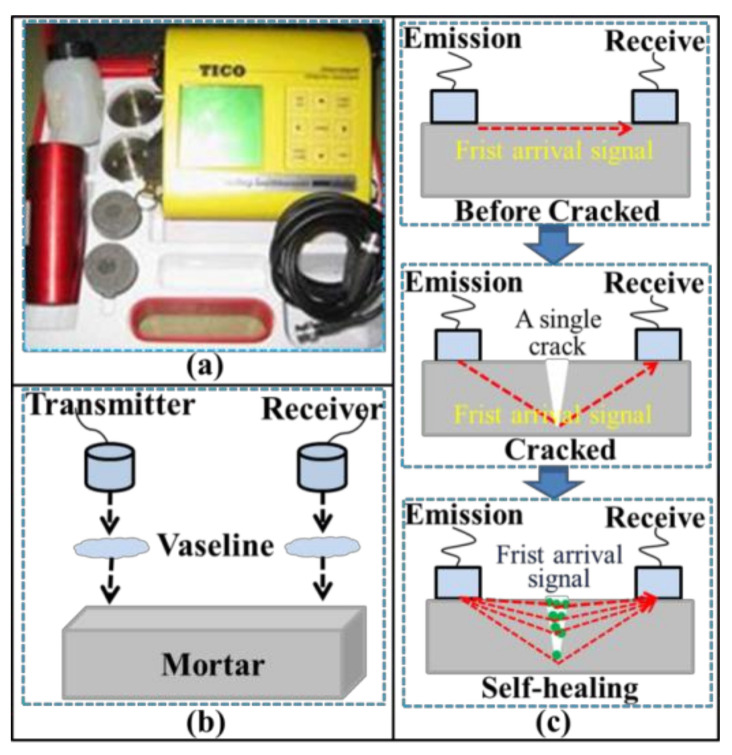
Schematic diagram of UPV test: (**a**) equipment; (**b**) sensor location; and (**c**) the mechanism of UPV test.

**Figure 4 materials-14-02068-f004:**
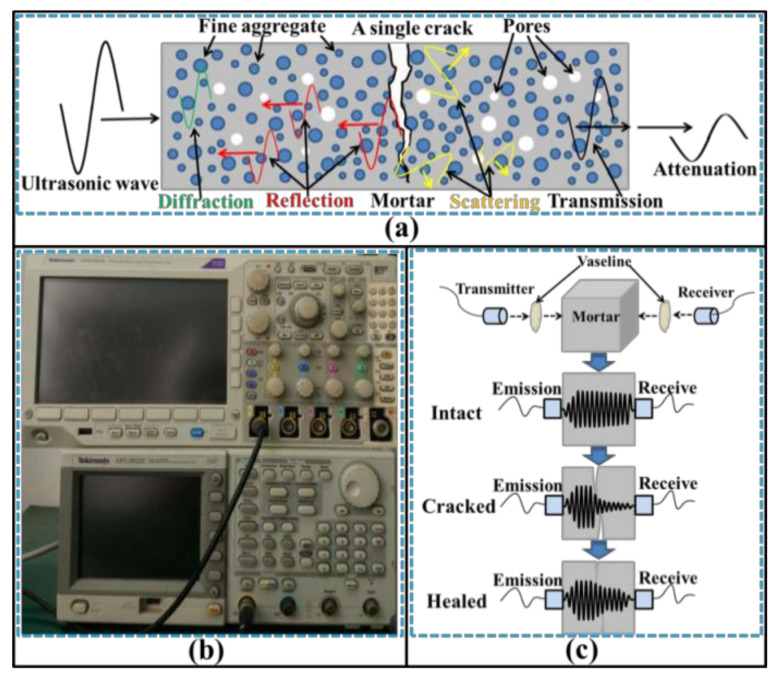
Schematic diagram of ultrasonic wave test: (**a**) the transmission mechanism of ultrasonic wave in concrete, (**b**) equipment, and (**c**) the wave attenuation.

**Figure 5 materials-14-02068-f005:**
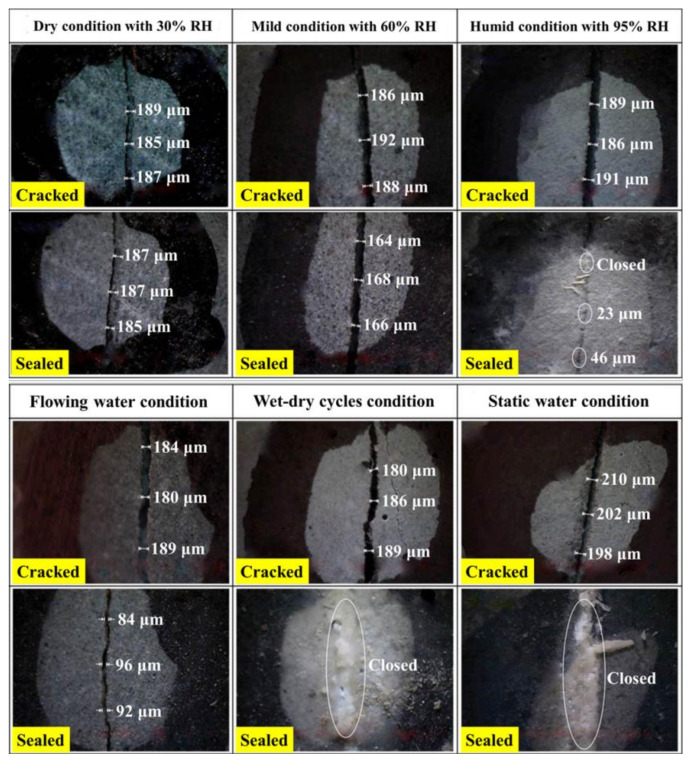
Surface cracks closure for the samples cured in different conditions.

**Figure 6 materials-14-02068-f006:**
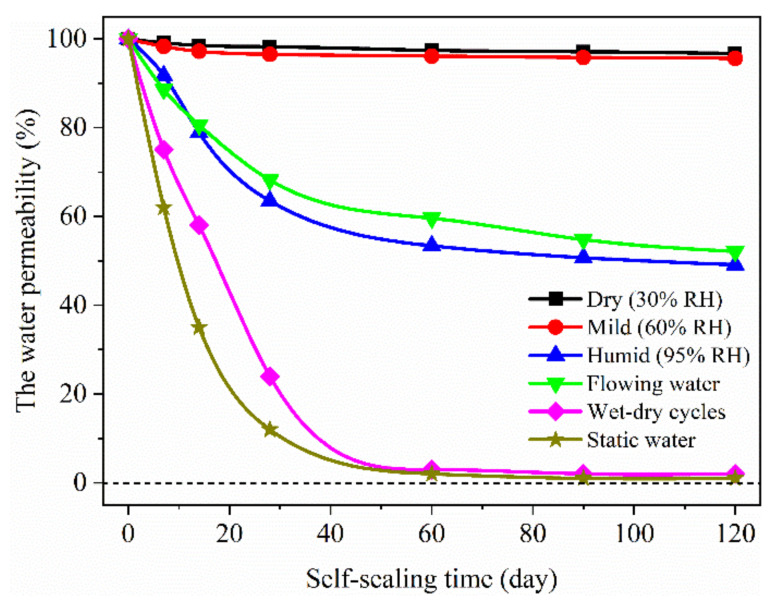
Water permeability for the samples cured in different self-sealing conditions.

**Figure 7 materials-14-02068-f007:**
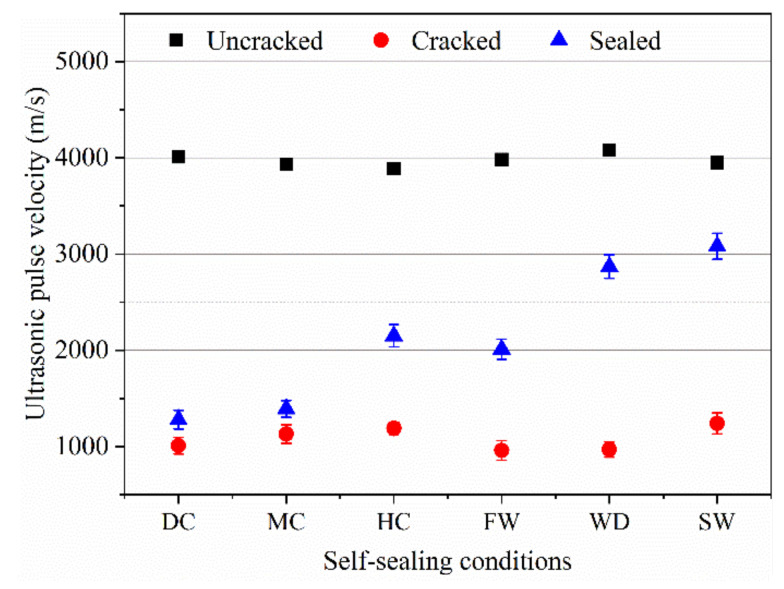
UPV for the samples with different states (DC: dry condition with 30% RH, MC: mild condition with 60% RH, HC: humid condition with >95% RH, FW: flowing water condition, WD: wet-dry cycles condition, SW: static water condition. The abbreviations are the same for the figures below).

**Figure 8 materials-14-02068-f008:**
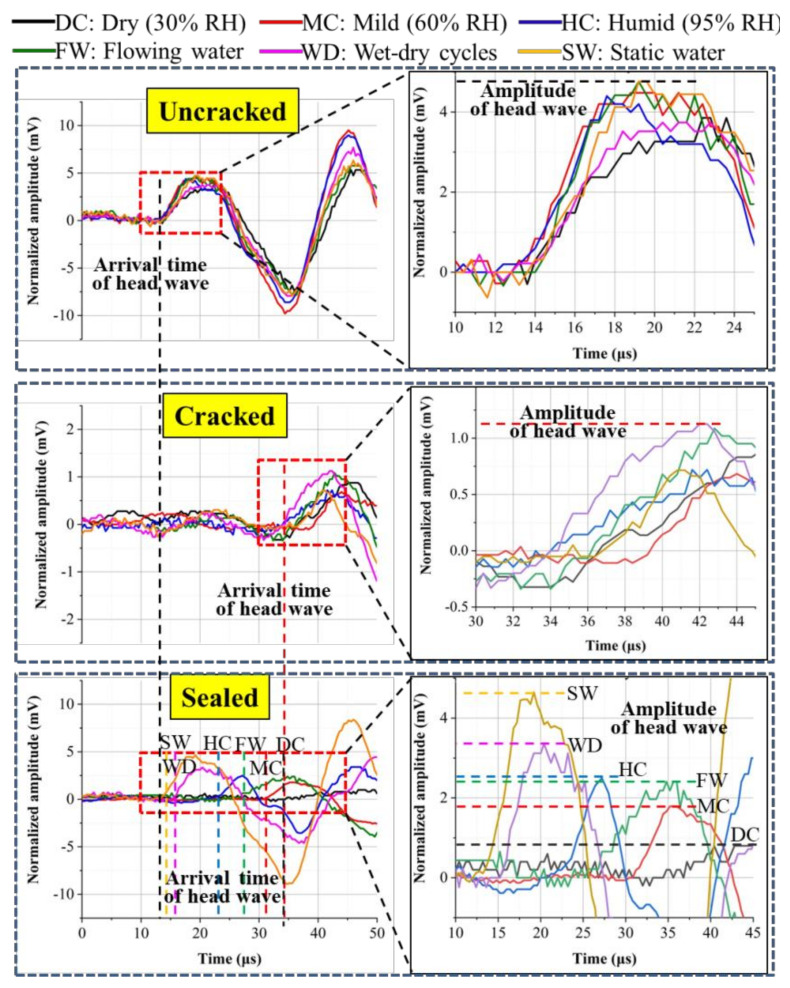
Arrival time and amplitude of head waves for the samples cured in different conditions.

**Figure 9 materials-14-02068-f009:**
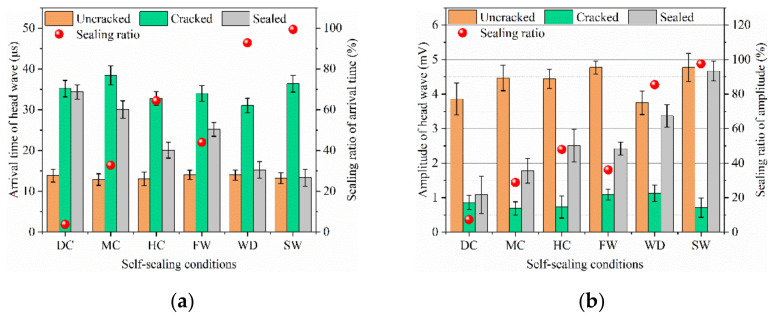
Self-sealing ratios based on the arrival time (**a**) and amplitude (**b**) of head waves.

**Figure 10 materials-14-02068-f010:**
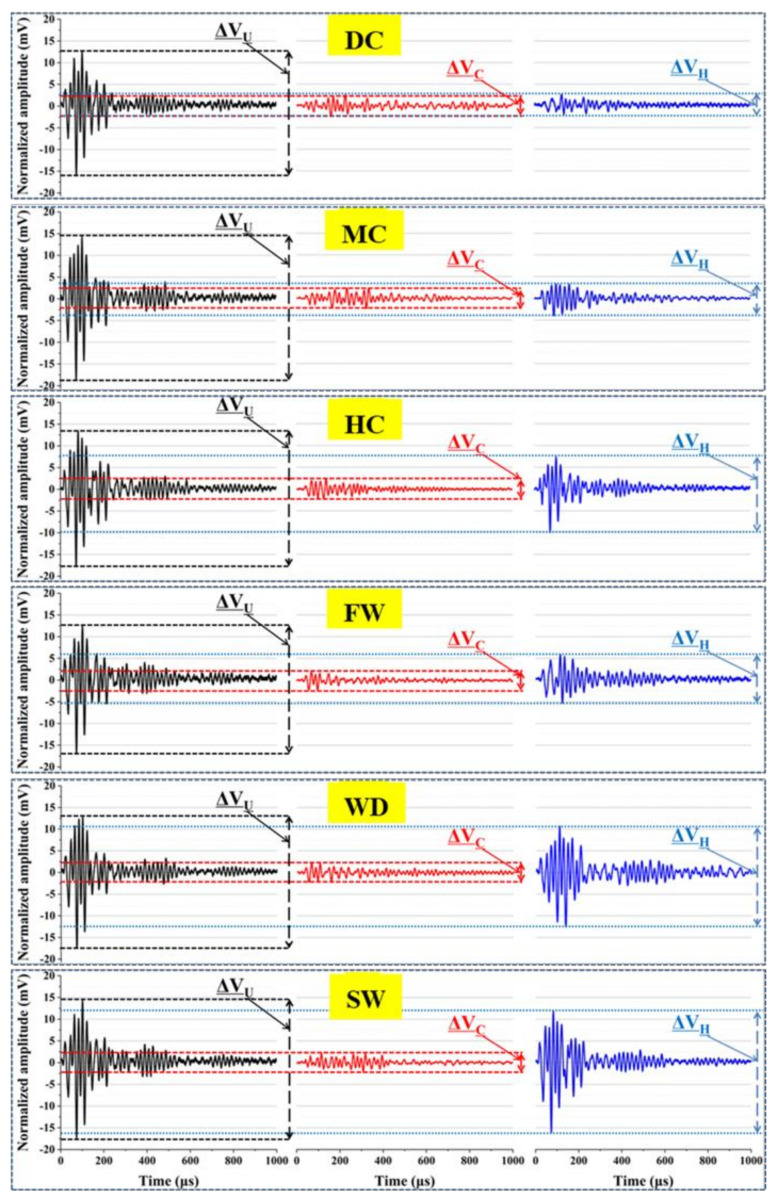
Ultrasonic waveform of samples cured in different conditions.

**Figure 11 materials-14-02068-f011:**
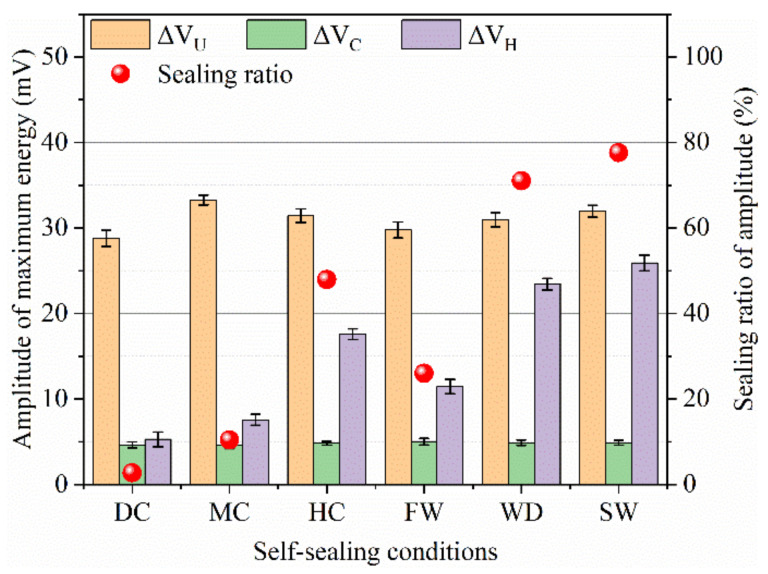
Self-sealing ratios based on the maximum amplitude of ultrasonic waves.

**Figure 12 materials-14-02068-f012:**
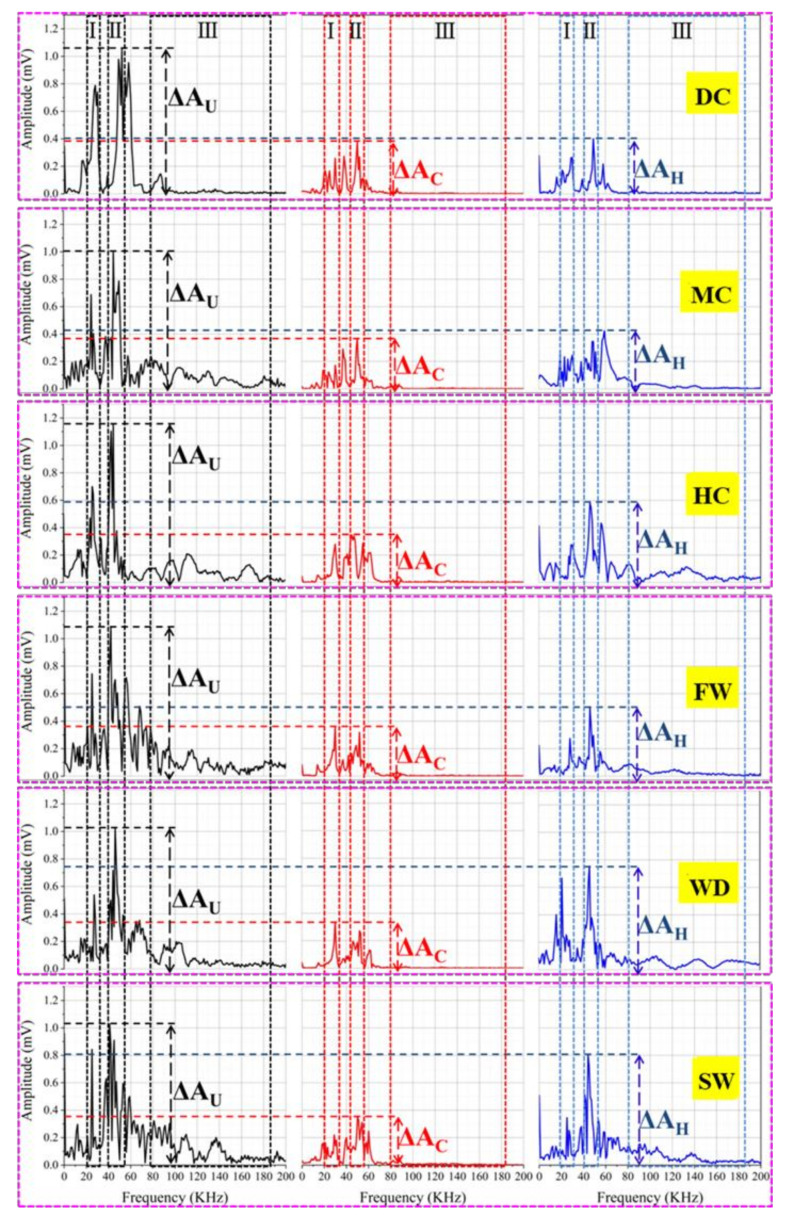
Frequency of ultrasonic waves for the samples with different states: the black, red, and blue lines belong to the uncracked, cracked, and sealed samples, respectively.

**Figure 13 materials-14-02068-f013:**
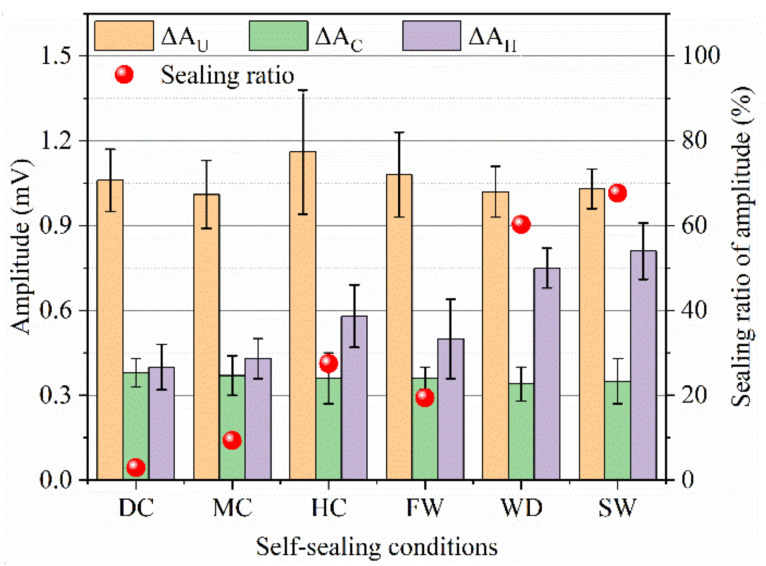
Self-sealing ratio based on the amplitude of dominant frequency.

**Figure 14 materials-14-02068-f014:**
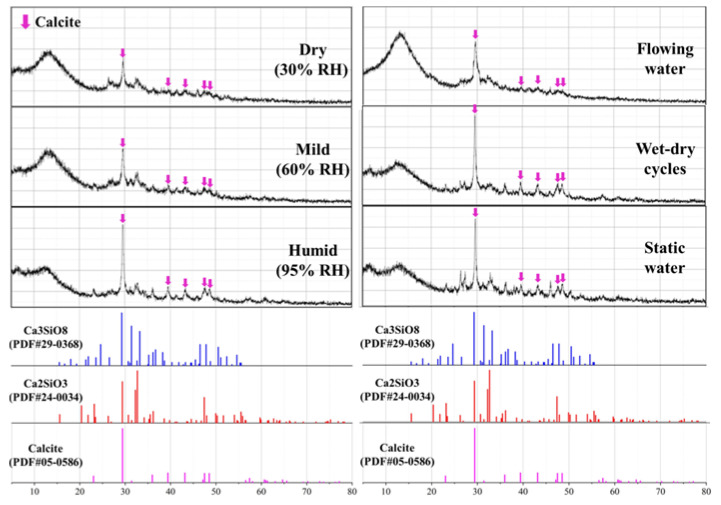
XRD of self-sealing products for the samples cured in different conditions.

**Figure 15 materials-14-02068-f015:**
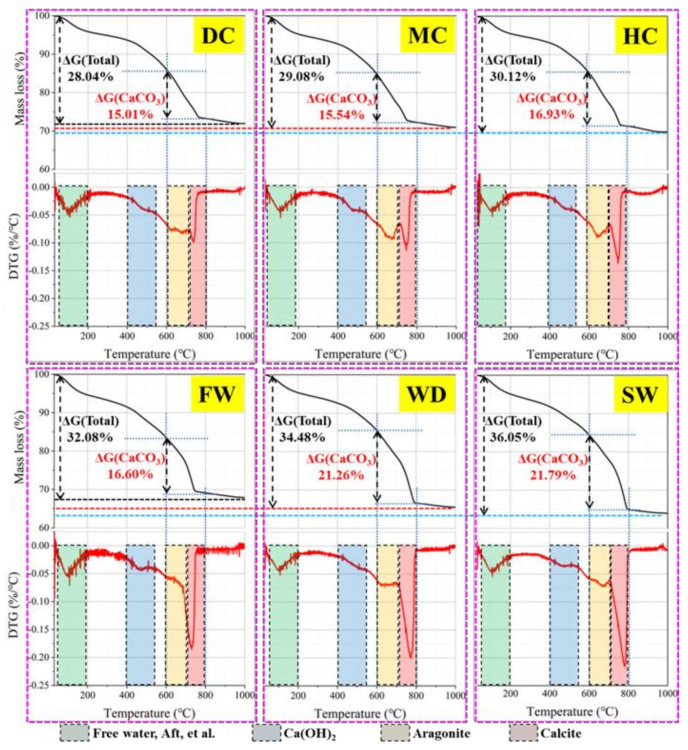
TG analysis of self-sealing products for samples cured in different conditions.

**Figure 16 materials-14-02068-f016:**
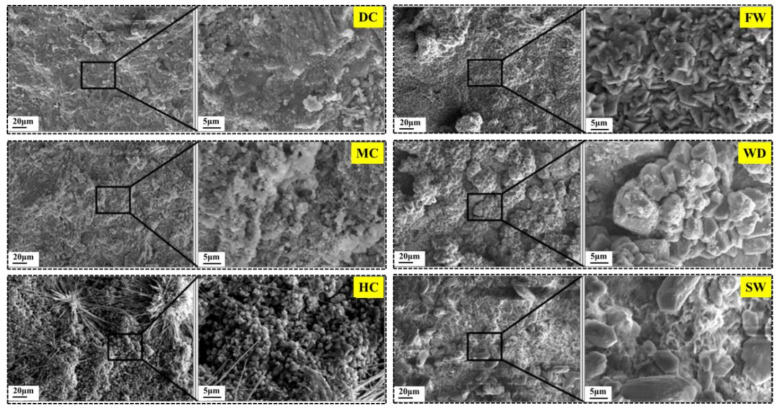
SEM of self-sealing product for samples cured in different conditions.

**Figure 17 materials-14-02068-f017:**
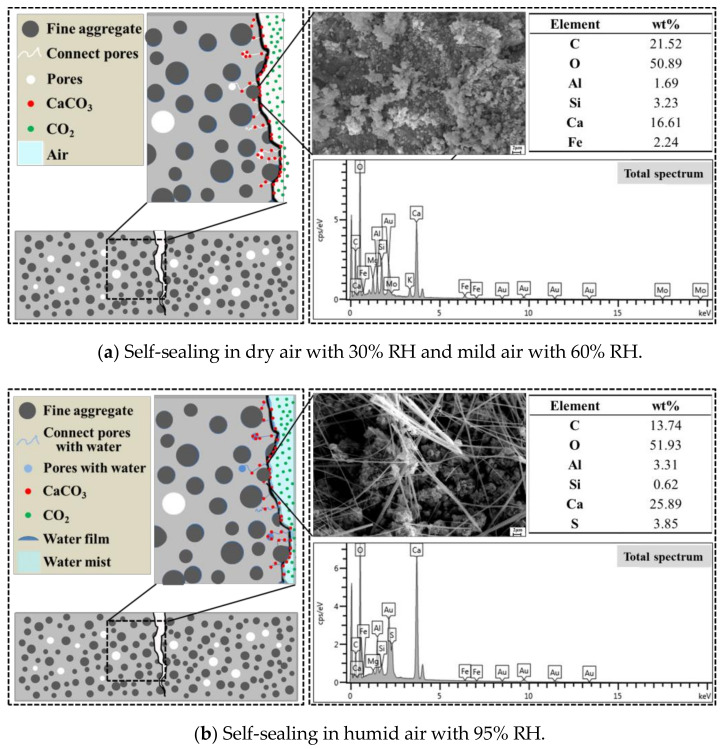
Mechanisms of self-sealing for samples cured in different conditions.

**Table 1 materials-14-02068-t001:** Chemical compositions of OPC (%).

Compositions	CaO	SiO_2_	Al_2_O_3_	MgO	Fe_2_O_3_	SO_3_	Others
Content (%)	57.82	22.16	6.36	2.94	2.78	4.20	3.74

**Table 2 materials-14-02068-t002:** Physical properties of OPC.

Surface Area (m^2^/kg)	Average Particle Size (μm)	Setting Time (min)		Compressive Strength (MPa)	
		Initial	Final	3 d	28 d
346	18.85	80	220	23.4	48.7

## Data Availability

The data presented in this study are available on request from the corresponding author.

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
