# Peer review of "Improving the Autogenous Self-Sealing of Mortar: Influence of Curing Condition"

_materials, 2021, doi:10.3390/ma14082068_

Round 1
Reviewer 1 Report
This paper deals with an interesting subject: the influence of curing conditions on the autogenous self-healing of mortar. The results are interesting and well commented except in some cases.
Besides, I do not have the feeling to recommend it for publication in its current form.
I would re-consider the paper once the authors have properly addressed the following issues:
Fig. 1 is not crucial for the purpose of the paper, therefore it can be eliminated.
In section 2.1 authors should add some information regarding the sand, for example the maximum diameter, the particle size distribution, the type of sand.
In section 2.3.1 the authors are advised to improve the explanation of Fig. 2 also in the text for the different specimens.
Also in section 2.3.3 the explanation must be improved: Mh is the mass of passed water per minute of the sample after making the crack, while Mc is the mass of passed water per minute of the same sample after the curing period for the self-healing process?
How many days of curing does the evaluation of UPV reported in Section 3.1 for self-healed samples refer to? Does it refer to 120 days? This information must be added by the authors.
From figure 7 it is possible to deduce that the self-healing process is faster in case of static water than wet-dry cycles. Also the speed of self-healing is an important parameter and this is not highlighted by the authors in the text.
The authors should correct Fig.18 (translate all in English).
The authors should correct some small typos, such us in line 277, and improve exposure in some sentences.
Author Response
Dear Reviewer,
Thank you very much for your kind letter and the reviewer comments. Those comments are all valuable and very helpful for revising and improving our paper to match the high standard set by the journal. Based on the suggestions and comments, we have made relevant amendments. We strongly believe that the revised manuscript is now much better than the previous one, both from a technical and literacy point of view and can now meet the high standard of the journal.
Please see the attachment.

Reviewer 2 Report
The paper describes an interesting issue of the mortar self-sealing. The paper contains the results of non-destructive testing methods of small mortar samples, which were cracked and cured in strictly defined conditions.
The introduction is interesting and explains the planned conditions for the curing of the samples.
The part concerning the research results is described in sufficient detail, except for the ultrasonic tests that are even too extensive presented, which is unnecessary and does not bring any new information.
The article lacks a substantive (qualitative) discussion part on the self-sealing process and some theoretical description of the phenomena taking place there. The discussion of the results should also be supplemented with a reference to the results of research made by other authors.
The conclusions presented are too general.
Detail suggestions and requirements
- The subject discussed in this paper is self-sealing of the mortar and this term should be used in relation to the described issue. Self-healing is associated with the use of additional substances / bacteria that improve the condition of the element over time. Please use this term - self-sealing - throughout the paper, including the title.
- Lack of empty line above and below figures / tables causes high illegibility of the information.
Point 2
- Does the OPC used in this study differ from the typically used material? Please refer to this in the context of other studies.
- How many samples were made for each type of test? Which type of samples were used in the described laboratory tests? Material tests require the average value and standard deviation to be provided – please complete these values in results.
- The preparation method of the cracks should be described in more details, the role of the epoxy resin and how it was remove from the sample surface before testing.
- Fig. 2 – there is no description for (a), (b) and (c)
- Ultrasonic test is an old and well-known technique for assessing the technical state of concrete by non-destructive method; making this test “used for the evaluation of concrete self-healing in recent years” is an abuse.
Point 3
- Figs: 8, 10, 12, 14 - please do not connect the points horizontally, because each dot represents a different environment, so there is no correlation here.
- 10 – showing percentages above 100% (Healing ratio) does not make sense.
- All the studies carried out using UVC (point 3.3) show identical correlations between self-healing and the environment in which the samples were cured. This is obvious because it results from measuring (only in different ways) identical values (amplitude, frequency, etc.). From the practical point of view describing so many methods of processing the ultrasonic results is unnecessary, because obtained relationships must be identical.
- XRD analysis (point 3.4.1) – there is no description of the results shown in the figure 15.
Author Response

(The authors gave the same response as above.)

Round 2
Reviewer 2 Report
I would like to thank the authors for the corrections and additions introduced. In this article, I miss a theoretical approach to the observed processes, but in its present form, the article is suitable for publication.